# Impact of pulse oximetry on hospital referral acceptance in children under 5 with severe pneumonia in rural Pakistan (district Jamshoro): protocol for a cluster randomised trial

Fatima Mir,[1] Apsara Ali Nathwani,[1] Suhail Chanar,[1] Amjad Hussain,[1] Arjumand Rizvi,[1] Imran Ahmed,[1] Zahid Ali Memon,[2] Atif Habib,[1] Sajid Soofi,[1] Zulfiqar Ahmed Bhutta[3]

SS and ZAB are joint senior authors.

¹Department of Paediatrics & Child Health, Aga Khan University, Karachi, Sindh, Pakistan
²Centre of Excellence in Women and Child Health, Aga Khan University, Karachi, Sindh, Pakistan
³Division of Women and Child Health, Aga Khan University, Karachi, Pakistan

**Correspondence to**
Dr Fatima Mir;
fatima.mir@aku.edu

## ABSTRACT

**Background** Pneumonia is a leading cause of death among children under 5 specifically in South Asia and sub-Saharan Africa. Hypoxaemia is a life-threatening complication among children under 5 with pneumonia. Hypoxaemia increases risk of mortality by 4.3 times in children with pneumonia than those without hypoxaemia. Prevalence of hypoxaemia varies with geography, altitude and severity (9%–39% Asia, 3%–10% African countries). In this protocol paper, we describe research methods for assessing impact of Lady Health Workers (LHWs) identifying hypoxaemia in children with signs of pneumonia during household visits on acceptance of hospital referral in district Jamshoro, Sindh.

**Methods and analysis** A cluster randomised controlled trial using pulse oximetry as intervention for children with severe pneumonia will be conducted in community settings. Children aged 0–59 months with signs of severe pneumonia will be recruited by LHWs during routine visits in both intervention and control arms after consent. Severe pneumonia will be defined as fast breathing and/ or chest in-drawing, and, one or more danger sign and/ or hypoxaemia (Sa02 <92%) in PO (intervention) group and fast breathing and/or chest in-drawing and one or more danger sign in clinical signs (control) group. Recruits in both groups will receive a stat dose of oral amoxicillin and referral to designated tertiary health facility. Analysis of variance will be used to compare baseline referral acceptance in both groups with that at end of study.

**Ethics and dissemination** Ethical approval was granted by the Ethics Review Committee of the Aga Khan University (4722-Ped-ERC-17), Karachi. Study results will be shared with relevant government and non-governmental organisations, presented at national and international research conferences and published in international peer-reviewed scientific journals.

**Trial registration number** NCT03588377.

## Strengths and limitations of this study

► Our study will directly assess impact of pulse oximetry on family referral acceptance by comparing family acceptance of hospital referral (verbal acceptance with hospital visit) in 'pulse oximetry and clinical signs' group with 'clinical signs alone' group.

► The study has applicability because it assesses impact with 'real-life' limitations (variability in lady health worker accuracy in identifying severe pneumonia, availability of oxygen and human resource at referral hospital and, availability of private transportation).

► The study design provides means of minimising the effect of confounding.

► The study design avoids bias in allocation to exposure groups.

► Blinding is not possible due to nature of intervention.

► Some subjects may fail to adhere to protocol and non-adherence may cause an underestimated measure of association.

## INTRODUCTION

Pneumonia accounts for an estimated 18% of under 5 mortality across the globe.[1] Majority of these pneumonia-specific deaths occur in 15 countries, in which Pakistan ranks fifth.[2] Failure to seek early care and delays in hospital referral are commonly acknowledged determinants of mortality in childhood pneumonia with a higher proportion reported from rural settings than urban.[3–6] Acceptance rates of 'facilitated' hospital referral advice have been reported low between 8% and 23% for sick young infants in periurban Karachi.[7 8] They are even lower for non-facilitated referral in rural settings in children under 5 with severe pneumonia in rural Matiari district, Sindh.[5] The prominent reasons in low-income and middle-income countries behind this delay are inability to recognise seriousness of pneumonia, distance from health facility and lack of money for private healthcare.[9 10]

In 1994, the government of Pakistan introduced the Lady Health Worker (LHW) programme in rural populations with low physician density to address common health problems in women and children under 5 through household visits. To date, a team of over 110 000 LHWs are working for the programme nationwide with 23 185 LHWs in Sindh alone.[11] Each LHW is responsible for a population of 1000–1500 individuals (catchment of about 100 families). Recruits are preferably local, with a minimum of 8 years of formal schooling, followed by 15 months of training to deliver maternal and child healthcare (MCH) in community settings. During a monthly home visit, the LHW provides essential MCH services including family planning needs, nutritional assessments of both mother and child, management of minor and common illnesses, improving immunisation coverage and imparting health education.[12]

This study recognises the LHW as a powerful conduit for reaching under 5 children at household level and identifying severe pneumonia in under five at an early stage. It also recognises the underestimation of hypoxaemia (arterial oxygen saturation, $SpO_2$ of <90%), a major risk factor for pneumonia mortality at community level and assesses its prevalence. The reported prevalence of hypoxaemia in under 5 acutely ill children is 5%–58% in facilities[13–16] and 16%–39% in community settings.[17] A 4.3-time higher risk of mortality has been associated with pneumonia with hypoxaemia than in children with pneumonia without hypoxaemia.[13] Hypoxaemia is also predictive of treatment failure with amoxicillin in children aged 3–59 months.[18] In resource-poor settings where pulse oximetry is not feasible, signs and symptoms of severe pneumonia (sleepiness, cyanosis, head nodding/grunting and inability to move) are used as predictors of hypoxaemia. However, the validity of clinical signs to predict hypoxaemia varies and it is often difficult for physicians working in settings, where objective detection of hypoxaemia is not available, to decide whether the child coming with severe pneumonia requires administration of oxygen or not.[15 19 20]

Pulse oximetry[16] is a rapid, portable, non-invasive and accurate method of measuring $SpO_2$ and has therefore been used in trial and clinical settings to detect hypoxaemia. Appropriate oxygen therapy (based on PO findings rather than clinical signs of severity alone) has been associated with lower mortality risk.[19] Assuming access to supplemental oxygen, PO could potentially avert up to 148 000 severe pneumonia-related deaths if implemented, and, combining PO with IMCI assessment for pneumonia has been shown to be cost-effective in 15 high burden countries.[21] Emdin *et al* found first level LHWs in periurban Karachi could easily perform pulse oximetry on young infants on well and sick visits to a primary healthcare facility.[22]

Over the past decade, the possible impact of pulse oximetry in hospital and community settings has been of interest across the globe. Health survey of 54 countries in 2010, suggested that 19.2% of the operating theatres around the globe are not equipped with pulse oximeters.[23] Trials assessing utility of pulse oximetry on a health systems level in Nigeria have shown that health workers reserve PO for the sickest patients.[24 25] This has also been observed in areas at higher altitudes with higher prevalence of hypoxaemia (highlands of Papua New Guinea).[25 26] There is lack of clarity about how pulse oximetry can be used in the community where lower hypoxaemia prevalence may be a lesser incentive for health workers to use PO such as in interior Sindh settings in Pakistan. Then again, health workers recruited from within communities may be better invested and motivated than hospital personnel in following case management guidelines precluding PO. It is, therefore, worthwhile to explore and describe contexts behind a family's acceptance (or not) of referral advice whether based on technology and/or clinical examination in rural settings where the highest burden of pneumonia deaths lies.

Feasibility and sustainability audits of oxygen delivery systems in the Gambia and Egypt have shown that providing technology alone is ineffective, and should preclude provision of supplies, education, training and feedback.[27–30] This protocol paper describes a study to assess the effect of PO monitoring in community settings on hospital referral acceptance in children under 5 with severe pneumonia.

## METHODS

### Study aims and design

The overall aim of the study is to assess if detection of hypoxaemia, and/or severe pneumonia in children 0–59 months by LHWs during their monthly home visits will increase hospital referral acceptance among families in district Jamshoro, Sindh, Pakistan.

Primary objectives are:

1. To assess and compare the impact of 'pulse oximetry' used by LHWs at household level on increasing hospital referral acceptance rates in intervention clusters (district Jamshoro) for children aged 0–59 months with severe pneumonia with the impact of LHWs using clinical signs alone in non-intervention clusters of the same district.
2. To investigate the likely predictors (demographic, clinical) of hospital referral acceptance in both the groups.

Secondary objective is:

3. To compare clinical outcomes (treatment completion, treatment failure, hypoxaemia) of children 0–59 months who accepted referral to those who refused admission and were treated at home.

A community-based cluster randomised trial will be conducted in district Jamshoro, among children of ages 0–59 months (figure 1). In intervention areas, all children with cough, fever or difficulty in breathing (acute respiratory illness) will be assessed for study eligibility (signs and symptoms of severe pneumonia, or hypoxaemia alone, or severe pneumonia with hypoxaemia) by LHWs during their monthly home visit in their catchment area. In control areas, study eligibility will require detection of

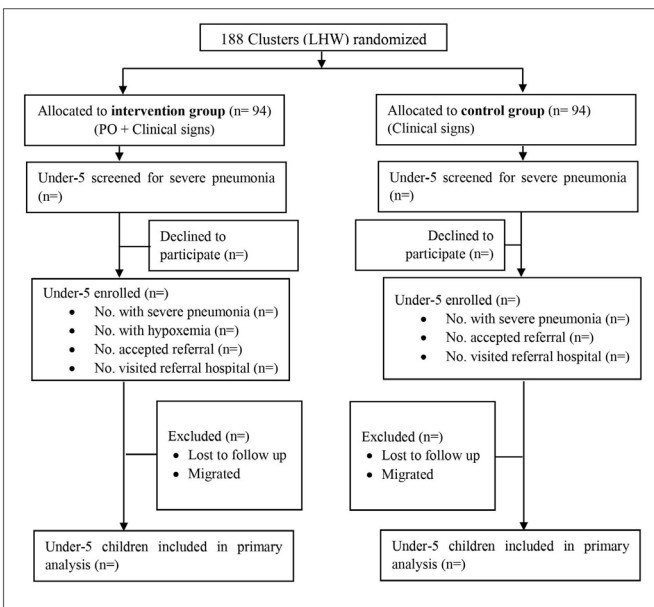

**Figure 1**  Trial profile. LHW, Lady Health Worker, PO, Pulse Oximetry.

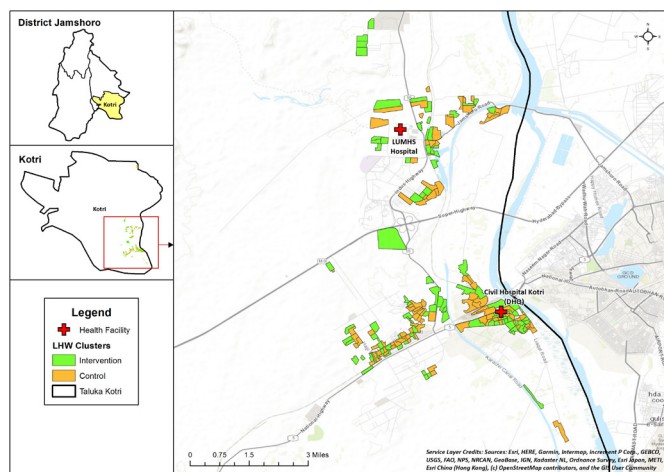

**Figure 2**  Map of intervention and control sites Abbreviations used in Map: Liaqat University of Medical and Health Sciences Jamshoro (LUMHS). Legend: Lady Health Worker (LHW). Service Layer Credits: Companies/institutes which contributed and collaborated to make this basemap publicly available (Environmental Systems Research Institute (ESRI), HERE (private company), General Bathymetric Chart of the Oceans (GEBCO), US Geological Survey (USGS), Food and Agriculture (FAO), National Park Service (NPS), National Resources Canada (NRCan), GeoBase, Institut Geographique National (IGN), Ministry of Economy, Trade and Industry (METI).

signs of severe pneumonia alone (online supplemental appendix 1) during LHW monthly visits. Data will be collected on demographics, likely predictors and clinical outcomes using a structured questionnaire.

The primary outcome is hospital referral acceptance in children under 5 with severe pneumonia and to identify demographic and clinical predictors of hospital referral acceptance. The predictors will include distance of child home to referral facility, socioeconomic status of household, parental education, child age, nutritional status, respiratory rate, temperature, hypoxaemia and presence of other illnesses.

Secondary outcomes include duration of oxygen therapy, treatment failure, duration of hospital stay, vital and health status of child at day 7th and 14th.

### Trial setting

The study will be conducted within the community of Taluka Kotri in District Jamshoro, Sindh (figure 2). Jamshoro District has a population of 993 142.[31] It is predominantly rural, with 33% literacy, and >50% employed daily wage labourers. A baseline survey conducted as a part of current study showed the status of overall health indicators: skilled birth attendance 57%, antenatal care coverage 75%, postnatal care cover for mother and newborn within 48 hours 31.5%, vaccination completeness in children 12–23 months 68% and care seeking for ARI and diarrhoea >80% (internal survey). A total of 27 health facilities function in the district including 1 district headquarter (DHQ) hospital, 3 taluka headquarter hospitals, 5 rural health centres and 18 basic health units. The district is divided administratively in 30 union councils. Kotri is one of the four Talukas of Jamshoro, consists of 44% (437 561) of the population of district.[31]

Participants will be the permanent residents of Kotri, Jamshoro and recruited from their homes during routine monthly visits by LHWs serving in their catchment areas. This study is expected to run for 48 months with participant identification and enrolment conducted simultaneously in intervention and control clusters over 21 months after an initial pilot of 1 month. Each enrolled child will be revisited at day 7 and 14 for outcome measurement.

### Participant

Any child aged 0–59 months having signs and symptoms of acute respiratory illness (cough, fever, difficulty in breathing) in intervention clusters will undergo assessment of (1) signs and symptoms of severe pneumonia and (2) pulse oximetry during monthly LHW home visits. Presence of severe pneumonia with or without hypoxaemia, or hypoxaemia alone will merit hospital referral (nonfacilitated). Any child aged 0–59 months having signs and symptoms of acute respiratory illness (cough, fever, difficulty in breathing) in control clusters will undergo assessment of (1) signs and symptoms of severe pneumonia alone. Presence of severe pneumonia will merit hospital referral (non-facilitated). Exclusion criteria will include lack of consent and, non-availability (lost to follow-up, migration) on days of scheduled follow-up visits (days 7 and 14). Those who accepted referral but did not show up at hospital will be included in final analysis.

### Randomisation and masking

Study clusters were defined as the area covered by an LHW. Each LHW covers a minimum of 100 households.

A list of LHWs working in Kotri was collected from the LHW programme, health department government of Sindh. There are a total of 188 active LHWs in the study site. A baseline survey was conducted to collect data on health indicators from the LHW catchments. The clusters were randomly allocated to intervention and control groups on 1:1 fashion with a computer-generated randomisation sequence that was generated by an independent expert. Clusters were matched on under 5 population and distance to referral health facility. No stratification was used for allocation; clusters were selected to ensure that the reporting and training centres of intervention and control LHWs were separate. The investigators and the national and provincial LHW programme coordinators will be excluded from the allocation process.

### Training of LHWs
The LHW programme of Pakistan consists of a community-based group of first-level health workers with the principal mandate of home-based maternal and child health. The recruitment process is well defined and selection criteria include: at least 8 years of education with middle school pass, local residency, recommendation from the community and preferably married.[32] Once selected, they receive 15 months of basic training in mid-wifery and family planning using standardised training manuals and curriculum, and periodic refresher training courses. Each of these LHWs is typically responsible for approximately 1000 people, or 150 homes, and often serve as the primary healthcare contact in these rural communities.[32 33]

We chose LHWs as study personnel due to their access to homes on regular monthly basis. We trained intervention and control LHWs in separate groups for all of the following: (1) classification of ARI (no pneumonia, pneumonia, severe pneumonia) using standard acute respiratory infections training modules (WHO and IMNCI), (2) identification of danger signs and (3) case management of pneumonia at home with oral amoxicillin and severe pneumonia with stat dose of antibiotic before hospital referral. Intervention LHWs received an additional training in using a pulse oximeter and obtaining a valid reading. The principal investigator (PI) led these training sessions with senior trainers of the LHW programme (lady health supervisors).

### Participant recruitment and study procedures
#### Intervention delivery
Children aged 0–59 months with cough and/or difficult breathing during regular home visits of LHWs will be assessed for first, signs and symptoms of severe pneumonia (fast breathing/chest in-drawing and one or more danger sign (unable to eat/drink, vomiting, convulsion and lethargy/unconsciousness) and/or stridor)) and second, hypoxaemia (SpO2 <92%) using a handheld pulse oximeter (Masimo Rad-5v) to measure blood oxygen saturation level. LHWs will also do case management of children with pneumonia and severe pneumonia. A 3-day course of oral amoxicillin will be given to children with

pneumonia at home, whereas children meeting referral criteria (severe pneumonia alone, hypoxaemia alone or severe pneumonia and hypoxaemia) after obtaining informed consent (online supplemental appendix 2), will be administered a stat dose of oral amoxicillin and referred to nearest referral hospital (DHQ Kotri).

The study investigators will have provided these pulse oximeters to the LHW programme in advance and highlighted which ones will receive them. Physicians at the referral centre serving the intervention clusters will also receive handheld pulse oximeters. All the LHWs and staff will be trained on the use, and maintenance of these pulse oximeters. Children with severe pneumonia with or without hypoxaemia will be advised to go to hospital for antibiotics and oxygen, using the PO reading as a tool to convince parents. Children with hypoxaemia alone, without signs of severe pneumonia, will be referred to hospital to rule out cyanotic congenital heart disease. Name of the predesignated health facility with available oxygen and study physician will be provided to all the LHWs so that Study Workers (non-LHW study personnel) can coordinate with study physicians and ensure the patient has reached and is receiving safe and recommended care at referral facility. Project staff will pretest and regularly monitor PO accuracy and quality of readings.

Hypoxaemia will be defined as an SpO2 <92%. SpO2 measurement will be recorded after 1 min of stable observation. If the SpO2 comes 92% or less, the child will first be assessed for nasal obstruction with readings repeated after applying nasal saline drops. If repeat reading shows hypoxaemia, the child will be referred to nearest designated referral hospital and admitted for oxygen via nasal or nasopharyngeal route and intravenous antibiotics, as per recommendations.

### Implementation of active control: clinical signs assessment
Children aged 0–59 months with cough and/or difficult breathing during regular home visit will be assessed by LHWs for signs and symptoms of severe pneumonia (fast breathing/chest in-drawing and one or more danger sign (unable to eat/drink, vomiting, convulsion and lethargy/unconsciousness) and/or stridor)). A 3-day course of oral amoxicillin will be given to children with pneumonia at home, whereas children with severe pneumonia (eligible for recruitment) will be requested for informed consent and offered stat dose of oral amoxicillin and referral to nearest referral hospital.

### Procedure at referral facility
Children who accept hospital referral in both intervention and control clusters and reach hospital premises with LHW referral slip will be assessed by study physician at the referral centre. An SMS notification with brief details of referred child will have been provided to trained study personnel (study physician) in advance at time of referral at both the referral facilities. Children with severe pneumonia and/or hypoxaemia as per LHWs

who reach referral hospital premises will be examined and subjected to pulse oximetry again by the study physician at referral facility. If signs and symptoms of severe pneumonia are present, the child will be admitted for further appropriate treatment (oxygen therapy via nasal or nasopharyngeal route and intravenous antibiotics, etc) and if the symptoms are not severe (absence of danger sign), the child will be treated in outpatient care as per the standard of referral facility. All the children admitted at referral facility will undergo 12 hourly monitoring by study personnel and filling of CRF and hospital physician form (HPF) at days 1, 7 and 14. Those children who refused the referral will be visited by study community health workers after 24 hours to confirm referral refusal and to fill CRF.

Preliminary meetings will be held with the executive director health Jamshoro, director general health Sindh, in-charge LHW programme Sindh and in-charge paediatric units LUMHS to ensure their cooperation through study duration. Emergency and paediatric unit staff at the referral facilities along with study personnel (physician/nurse) will be trained on management of severe pneumonia according to the integrated management of neonatal and childhood illnesses (IMNCI) guidelines.[34] A baseline survey will be conducted at the health facilities to ensure availability of oxygen and necessary intravenous antibiotics. Even though it is ideal to guarantee sustainable oxygen systems at the two chosen referral public sector hospitals, this study does not provide oxygen, and therefore, aims to assess 'real-life' situations in public hospitals and their impact on severe pneumonia outcomes with or without hypoxaemia. LHWs will be incentivised on basis of their contribution to the study activities.

### Data collection and storage

Data will be collected by LHWs during house visits (screening form), community health workers (during follow-up visits days 1, 7 and 14) and hospital-based study personnel (for all who accept referral and reach hospital premises) on paper forms (online supplemental appendix 3). Given that it will be a new experience for LHWs to assess, classify and manage ARI cases and at the same time record findings on data forms accurately, these will be supervised closely and frequently, at least for the first pneumonia season. Well trained study field supervisory officers and LHW supervisors will be required to perform regular field supervision in their respective clusters and ensure accurate and logically entered data forms and make necessary verifications and corrections at the data collection sites and give feedback to the LHW to avoid repeating the errors. Raw data brought to the programme office will be checked once again for accuracy by the technical staff and approved for entry in the computer. All raw data will be safely kept in the AKU office, appropriately numbered by cluster, until 7 years after the study is over.

### Case history records

These include the study CRF and HPF that will contain information that documents the child's eligibility to participate in the study, the signed consent form and information from tests and examinations. Wherever possible copies of supporting documentation for the information contained in the CRF should be kept with each patient's case history record. This supporting documentation may include records of physical examinations, progress notes, laboratory reports, X-rays, consultations, correspondence, information and data on the subject's condition, during and after the clinical investigation, diagnoses made, concomitant therapy, etc. All information in the case history records should be attributable to a specific individual. Since the CRF will not contain the patient's name, there will be a unique link between the ID number on the CRF and the patient's name. Each child's case history record will be evaluated to verify validity and completeness of the data on the CRF when a study monitor visits the study site. All corrections to CRF's must be made without obscuring the original entry. The revised entry should be inserted and the person making the correction should sign and date the correction. Only authorised study personnel may complete or correct CRFs.

### Data management

Screening data will be collected on paper by LHWs. CRF and HPF will be collected on electronic forms. To ensure proper implementation of the intervention, the field supervisors will make spot checks and will arrange monthly refresher group sessions of the first-line health workers in which the problems encountered will be discussed and resolved. In addition, the data collection activity will be carried out by teams consisting of LHWs/CHWs and study staff will be further monitored by field supervisors who will perform a check on a subset (5%) of households.

An information system will be set up to keep track all patients screened and enrolled and a filing system to keep all study related records—case history records, study protocol or related documentation and drug distribution records. The coordinator at the site will be responsible for the completeness and accuracy of all the study materials.

### Study protocol and related documentation

All study-related documents including the study protocol, manuals of operations, all correspondence sent to or received from the study monitor, materials used for obtaining informed consent, protocol modifications and records of the institutional review board (IRB) approval and all communications with the IRB must be maintained in complete form. These documents will be evaluated to ensure that study documentation is complete and current when a study monitor visits the study site.

### Record retention

Retention of accurate and complete records is essential to establish the validity and completeness of the study. All records must be retained for 7 years after the data set is frozen. Electronic data will be deidentified, unlinked from any personal identifiers and therefore will protect individual identity.

### Reporting of serious adverse events and treatment failures

Amoxicillin is in widespread use and is not investigational in any study site. However, since oral amoxicillin is not routinely recommended for initial treatment of children who have severe pneumonia, the appropriate case report forms (CRFs) describing the occurrence of a serious adverse event, treatment failure or death must be faxed to the coordinating centre within 72 hours of the site coordinator knowing about the event. Adverse events, treatment failure and death must be reported to allow appropriate interpretation of this critical information. If the outcome of the adverse event is unknown when the site coordinator first notifies the coordinating centre, a follow-up form must be faxed to the coordinating centre within 10 days of knowing about the event. The PI should send a copy of the adverse event data to their local IRB as soon as possible. The coordinating centre will summarise the adverse event and death information and send a report to the IRB/ERC of sponsors and to site IRBs. Both the rate of adverse events and the rate of patient accrual at each individual site will be monitored to determine if stopping rules are met. We do not anticipate serious adverse events. However, in case of one, a DSMB will be requested for and convened on ad hoc basis for safety review at any time during the study if there is a concern regarding rates of adverse events or rates of patient accrual. Adverse events will be reported by the study physician to the PI and clinically managed by the study physician in conjunction with other physicians at the institution. Any related and unexpected life-threatening adverse event including death will be reported to the IRB within two business days as per IRB protocol and any related, unexpected and serious adverse event will be reported to the IRB within 10 business days as per IRB protocol.

### Compliance with and deviations from the study protocol

The site coordinator must agree with and sign the protocol and confirm in writing that he or she has read, understands and will work according to the protocol and Good Clinical Practice. The site coordinator is responsible for making sure that the protocol is strictly followed and should not make any changes to the study unless necessary to eliminate an apparent immediate hazard or damage to a trial subject. Any deviations from the study protocol including but not limited to inappropriate enrolment of a study subject, administration of the wrong study treatment, missed doses of study treatment, missed observation points, incorrect administration of concomitant medications, etc should be reported to the coordinating centre and each site's IRB. The report should include a plan to rectify any problems at the site that may have caused the protocol deviation.

### Sample size

Defining a cluster as (the catchment area of) one LHW, and assuming a power of 90% in detecting 50% increase in referral acceptance from a 10% baseline to 15% among children aged 0–59 months with severe pneumonia (pneumonia prevalence at 2-week recall (MICS Sindh): 7.5% (18% of which is assumed severe pneumonia) with ICC 0.001736. We need to capture a total of 4160 children with severe pneumonia in both intervention and control groups.

### Data analysis

The primary analysis for each outcome will perform on an 'intention-to-treat' basis, that is, all children included in the analysis who were enrolled in the study according to the group to which they were allocated. All analyses will account for the cluster randomised design to ensure correct type I error rates and CIs.[35] Baseline characteristics will be compared by analysing differences in means and proportions among the study arms. Categorical outcomes will be compared using $\chi^2$ test and continuous outcomes using Student's t-test. For analysis of predictors of referral acceptance, generalised linear model will be used with logit link function. The univariate analysis will be conducted to explore the independent effect of each predictor on outcome. The variables significant at a liberal p <0.20 will be included in multivariate model for adjustment. The results will be reported as relative risk with 95% CI. Type 1 error will be set at 5% level. All analysis will be done using STATA V.15.

### Patient and public involvement

Patients or the public will not be involved in the design or conduct of the study. Results will be disseminated to the community.

### Study status

Recruitment began in August 2019 and field activities and data collection are in process. As of 16 October 2020, a total of 235 cases and 184 controls have been enrolled. Extension of study duration is in discussion to achieve sample size.

### DISCUSSION

Hypoxaemia, a frequent complication of severe pneumonia, is a major risk factor for death in children under 5. Theoretically, detection of hypoxaemia at community level among severe pneumonia cases by the use of pulse oximeter would give awareness to the caregivers about severity of illness and reduce delay in hospital referral. Delayed care-seeking is a recognised risk factor in pneumonia mortality in community settings.[36] Unfortunately, information on prevalence of hypoxaemia, effectiveness of its detection in influencing parents to seek hospital care (gold standard for severe pneumonia) and impact

of appropriate care at hospital in settings like Pakistan is lacking. Thus, the findings of this study will build evidence for utility of providing front-line workers like LHWs with a tool to detect hypoxaemia if signs and symptoms of pneumonia are present. Interestingly various groups across the world are now espousing pulse oximetry with other strengths like detection of congenital heart disease[37 38] and newborn sepsis.[39]

## Limitations

Some union councils with poor LHW coverage were not included in this study. This was a compromise on generalisability in favour of feasibility. It may have led to exclusion of children whose referral patterns were important to gauge. We also did not offer facilitated referral in either arm. This may have affected referral acceptance in both groups, however, will allow assessment of real-life impact of transportation barriers.

Though we kept hospital referral (as recommended by WHO) for severe pneumonia as our standard of care, it was beyond the scope of this study to ensure that referral centres had sustainable oxygen systems/capacity for non-invasive ventilation in children with severe pneumonia, or readily available cardiac evaluation resources for children with hypoxaemia alone. Duke *et al* showed improved case fatality rates by providing oxygen concentrators and pulse oximeters at five hospitals in PNG along with protocols for use.[35] Lack of quality care at referral hospitals in developing countries is a recognised barrier to healthcare seeking behaviour.[40 41]

If our study reveals pulse oximetry has influenced health-seeking behaviour significantly, we will need to follow with a more systematic evaluation of pneumonia care at rural hospitals which vary in quality of care. We will also need to correlate recovery rates in those who accepted hospital referral versus those who stayed home on oral amoxicillin. Evidence to support home care for severe pneumonia is poor.[42]

The COVID-19 pandemic has adversely affected recruitment in the study with disruption of field activities for many months in 2020. We anticipate prolongation of study duration in order to achieve sample size.

## ETHICS AND DISSEMINATION

This study has been approved by the Ethical Review Committee of The Aga Khan University (4722-Ped-ERC-17), Karachi, Pakistan in June 2017. Written informed consent in the local language will be obtained from parents or guardians of all participants. Data forms will contain no identifying information other than age, sex and GIS coordinates. Laboratory forms will contain no identifying information, specimens will be identified by a study number only and test results will not be linked to any individual by name. All survey staff will sign a confidentiality agreement to ensure that they do not release participant identities and test or study results to individuals who are not part of the study team.

Study progress and findings will be shared with sponsors (BMGF) quarterly. Results will be presented at national and international research meetings and conferences and also prepared for publication in international peer-reviewed scientific journals. Study findings will be disseminated to the study communities.

**Acknowledgements** We would like to acknowledge the mothers and families who have contributed to the study. We are grateful to the Sindh LHW Programme, Department of Health for their support and facilitation of the trial.

**Contributors** FM, ZAM, AH, SS and ZAB: conceptualisation of project. FM, AAN and SC: development of study design and questionnaires. AAN and SC: oversight of data collection. AAN, SC and AH: support of study logistics and field activities. IA and AR: statistical analyses. SS and ZAB: overall supervision and critical input. All authors have read and approved the final manuscript.

**Funding** The study is funded by Bill & Melinda Gates Foundation through grant OPP1148892.

**Competing interests** None declared.

**Patient consent for publication** Not required.

**Provenance and peer review** Not commissioned; externally peer reviewed.

**ORCID iDs**
Fatima Mir http://orcid.org/0000-0001-8602-5353
Apsara Ali Nathwani http://orcid.org/0000-0002-2899-1577
Suhail Chanar http://orcid.org/0000-0002-4087-1084
Sajid Soofi http://orcid.org/0000-0003-4192-8406

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
