## [Reviewer comments · BMJ Open]

ARTICLE DETAILS

TITLE (PROVISIONAL)	Impact of Pulse Oximetry on Hospital Referral Acceptance in Children under 5 with Severe Pneumonia in Rural Pakistan (District Jamshoro): Protocol for a Cluster Randomized Trial
AUTHORS	Mir, Fatima; Ali Nathwani, Apsara; chanar, suhail; Hussain, Amjad; Rizvi, Arjumand; Ahmed, Imran; Memon, Zahid Ali; Habib, Atif; Soofi, Sajid; Bhutta, Zulfiqar

VERSION 1 – REVIEW

REVIEWER	Arcott-Mills, Tonya The Children's Hospital of Philadelphia, General Pediatrics
REVIEW RETURNED	14-Jan-2021

GENERAL COMMENTS	Review: This is an important study and overall is well designed and I look forward to reading the results . The introduction and background are well written. The setting is clear. Case definition: How this is written needs to be more clear. As I understand it—a child must meet the clinical case definition of severe pneumonia and thus qualify for referral independently of the pulse ox measure. The pulse ox measurement is not actually a part of the case definition but an added measure in the intervention arm. I am assuming children who meet the case definition will still be referred in the intervention even if not hypoxic? What about those who are only hypoxic and not tachypneic or don't have a danger sign? Do they have to have both to get the pulse Ox measured? Make this more clear in text and appendix 1. Delivery of the arms—in the consent is clear that subjects will be reimbursed or provided with transport money if referred to the hospital. I am assuming this is independent of the arms and thus will not confound the results as it is the same in both arms. However, this should come out clearly in the text of the delivery of the arms. As cost to get to care can be a barrier any write up will also have to be clear that this was provided in both arms and thus the uptake overall could be higher than if this is not present and what should be focused on is the difference between the arms and not the absolute percentages. In the trial profile figure- why are you excluding those who don't show up at the hospital--- are they not a part of the outcome- those who do not accept referral? If they don't show even if they said they would go they are clearly indicating by the action that they are not accepting the referral. At least indicate clearly the rationale for this exclusion or a plan to analyze these as well. If they had consented and did not show would you not do the same CRF as those who refused referral? From the text I assumed that refused
--

	referral included those who said they were going and never arrived. Add to the participant section the exclusions after enrollment before analysis as well- those who do not show, lost to follow up, migrated. Also – more clearly outline the care at the referral center by what is standard vs trail clinician influenced. For example, what if on arrival at the center the center doctors do not detect hypoxia. What happens? What if they assess the child to not have severe pneumonia? Are they sent home? Admitted anyway? Who decides? Study Status/Recruitment—from the sample size calculation and the timeline of the trial and numbers recruited to date, recruitment seems to be far behind. Indicate how this will be addressed Data analysis- As I am not a statistician I cannot comment on the appropriateness of the analysis proposed.
--	--

REVIEWER	Lindtjørn , Bernt University of Bergen, Centre for International Health
REVIEW RETURNED	20-Jan-2021

GENERAL COMMENTS	This paper deals with an important issue. The study aims to use community health workers, called lady health workers, to identify hypoxemia in children with signs of pneumonia during household visits on acceptance of hospital referral in District Jamshoro, Sindh in Pakistan. The design is a community-based cluster randomized trial. The authors should write more about the lady health workers. Have they got any training? How are they followed up? Information in the paper states that the data collection has been completed. Nevertheless, when asking the editor, I understand that the trial is ongoing. The main outcome is «accepted hospital referrals» with severe pneumonia. However, the definition of what an accepted hospital referral is not clearly stated. The authors state that measuring the oxygen saturation with a pulse oximeter by the LHWs has been valuated. Even so, for people who are doing this in clinical or community settings, it is difficult to get reliable measurements in severely sick children who scream and are afraid. How is the research dealing with this challenge? On how many children did the LHWs fail to get a reliable measurement? Similarly, how are the authors validating the clinical features measured by the LHW? Is assessing the respiratory rate a straightforward exercise in a home setting in a crowded city? I would appreciate that the authors could explain the practicalities in doing this. I understand the wider aim of the study is to provide practical help in reducing child deaths. The external validity of this study will depend on how good the coverage of the LHWs is. Can the authors provide more information about this? Do the LHWs cover all the population? How many of the families would seek traditional medicine and how many would prefer the different private actors?
---

	The last objective is: To compare clinical outcomes (treatment completion, treatment failure, hypoxemia, duration of hospital) of children 0-59 months admitted with severe pneumonia who accepted hospital referral to those who refused admission and were treated at home. It is difficult to understand this objective as the outcome measure is dependent on hospital admission. It could be rephrased to compare the outcomes of those accepting referrals compared with those being managed elsewhere. I suspect that those being managed at home would also include children not receiving any treatment. Will there be repeated assessments in the homes after the initial contact?
--	--

VERSION 1 – AUTHOR RESPONSE

Reviewer1's comments:

1. Case definition: How this is written needs to be clearer. As I understand it—a child must meet the clinical case definition of severe pneumonia and thus qualify for referral independently of the pulse ox measure.

This is correct. In intervention arm, all children with history of acute respiratory illness (cough, fever, difficulty in breathing) will be assessed for signs for severe pneumonia and hypoxemia. Those who meet clinical definition of severe pneumonia will qualify for referral independent of pulse oximetry measure. Those who have hypoxemia alone will also qualify for hospital referral (Lines 155-159).

2. The pulse ox measurement is not actually a part of the case definition but an added measure in the intervention arm. I am assuming children who meet the case definition will still be referred in the intervention even if not hypoxic? What about those who are only hypoxic and not tachypneic or don't have a danger sign? Do they have to have both to get the pulse Ox measured? Make this clearer in text and appendix.

All children in intervention group with history of recent acute respiratory illness (cough, fever and difficulty in breathing) will undergo 1) pulse oximetry and 2) assessment of signs of severe pneumonia (respiratory rate measurement, chest indrawing and danger signs assessment). Either one or both together will qualify child for referral (Lines 155-159).

The term case definition has been changed to eligibility/referral criteria (Appendix 1).

3. Delivery of the arms—in the consent is clear that subjects will be reimbursed or provided with transport money if referred to the hospital. I am assuming this is independent of the arms and thus will not confound the results as it is the same in both arms. However, this should come out clearly in the text of the delivery of the arms. As cost to get to care can be a barrier any write up will also have to be clear that this was provided in both arms and thus the uptake overall could be higher than if this is not present and what should be focused on is the difference between the arms and not the absolute percentages.

Thank you for pointing this out. In the initial phases of the project, monetary incentivization versus facilitated referral had been considered. We found we had appended an initial version of the consent form. Eventually over the first year of discussing pros and cons of field implementation, the group decided in favour of real-life circumstances with non-facilitated referral in both arms (Lines 156, 159).

We will look at cost of transportation as one of the predictors of referral acceptance in both arms. We are appending the final version of consent form (Appendix 3).

4. In the trial profile figure- why are you excluding those who don't show up at the hospital--- are they not a part of the outcome- those who do not accept referral? If they don't show even if they said they would go they are clearly indicating by the action that they are not accepting the referral. At least indicate clearly the rationale for this exclusion or a plan to analyze these as well. If they had consented and did not show would you not do the same CRF as those who refused referral?

Thank you for this comment. We have corrected the flow chart and added those who reached hospital as a proportion of those who accepted referral verbally (Figure 1).

5. From the text I assumed that refused referral included those who said they were going and never arrived.

This is correct. We however agree with reviewer comment and will include these children with partial acceptance (verbal acceptance alone without reaching hospital) in analysis (Figure 1).

6. Add to the participant section the exclusions after enrolment before analysis as well- those who do not show, lost to follow up, migrated.

This has been added Lines 159-161. 'Who do not show' will not be excluded. They will be included in analysis and followed up for clinical outcomes through home visits (Lines 161-162).

7. Also – more clearly outline the care at the referral center by what is standard vs trial clinician influenced. For example, what if on arrival at the center the center doctors do not detect hypoxia. What happens? What if they assess the child to not have severe pneumonia? Are they sent home? Admitted anyway? Who decides?

A trained trial clinician was placed in both the referral facilities, with the responsibility of documenting hospital course for referred child who reached hospital premises. Management was done as per standard of referral centre (oxygen if available and intravenous ampicillin with escalation to ceftriaxone if no improvement in 48hours) for all children with severe pneumonia. We predicted that some of referrals from LHWs may not meet physician criteria of severity and allowed the physicians to use their discretion and discharge on oral antibiotics if appropriate. (Line 237-241)

8. Study Status/Recruitment—from the sample size calculation and the timeline of the trial and numbers recruited to date, recruitment seems to be far behind. Indicate how this will be addressed Data analysis- As I am not a statistician I cannot comment on the appropriateness of the analysis proposed.

The sample size is large and frequency of pneumonia is season-dependent with more cases in certain months (October-February) than others. The COVID pandemic and subsequent intermittent lockdowns, with intermittent POLIO vaccination campaigns (LHW dependent) slowed down field activities and recruitment. We are considering the option of no cost extension and are in discussion with our funders (Line 367).

9. The limitations could be more clearly laid out in a limitations section although the discussion does go into what this study will be able to answer and not...would be good to have a more described limitations section than what is briefly mentioned in the summary.

This has been added (Line 381 – 390, 400-402)

Reviewer 2's comments:

1. The study aims to use community health workers, called lady health workers, to identify hypoxemia in children with signs of pneumonia during household visits on acceptance of hospital referral in District Jamshoro, Sindh in Pakistan.

This has been done. (Line 105-107)

2. The design is a community-based cluster randomized trial.

This has been done. (Line 119).

3. The authors should write more about the lady health workers. Have they got any training? How are they followed up?

This has been added. (Line 175-191)

4. Information in the paper states that the data collection has been completed. Nevertheless, when asking the editor, I understand that the trial is ongoing. The main outcome is «accepted hospital referrals» with severe pneumonia. However, the definition of what an accepted hospital referral is not clearly stated.

The trial is ongoing. Our original end of trial date Dec 2021 may end up being extended.

Accepted hospital referral is verbal acceptance and showing up physically at the hospital. (Line 29)

5. The authors state that measuring the oxygen saturation with a pulse oximeter by the LHWs has been valuated. Even so, for people who are doing this in clinical or community settings, it is difficult to get reliable measurements in severely sick children who scream and are afraid. How is the research dealing with this challenge? On how many children did the LHWs fail to get a reliable measurement?

Our SOPs were designed to train to take two readings from each child in intervention clusters and take a second reading after decongestion of nose if concurrent upper respiratory symptoms. The Masimo Handheld pulse oximeters came with Velcro binders to minimize movement artifacts on sensor placement (finger or toe of child). The majority of LHWs have managed to get reliable readings.

We trained LHWs at the beginning and then subsequently at least 2 times during 2019 and 2020 pre-Covid. The LHWs were encouraged to self-report trouble with measurements. Some were detected during field supervision visits. Since April 2020, no group trainings were done but field supervisors did brief refreshers one to one with LHWs who had trouble with reliable readings and had low rates of referral.

6. Similarly, how are the authors validating the clinical features measured by the LHW? Is assessing the respiratory rate a straightforward exercise in a home setting in a crowded city? I would appreciate that the authors could explain the practicalities in doing this. I understand the wider aim of the study is to provide practical help in reducing child deaths.

Field monitoring was designed to catch a 10% sample of LHWs during each month and assess their clinical skills in detecting pneumonia. Most were responsible and efficient in RR measurement. We

found a tendency to over-read congestion sounds as stridor. This was mitigated through frequent refresher training on a one to one basis.

7. The external validity of this study will depend on how good the coverage of the LHWs is. Can the authors provide more information about this? Do the LHWs cover all the population? How many of the families would seek traditional medicine and how many would prefer the different private actors?

When planning the baseline survey, we found that only 88% of the district was covered by LHWs. Even though it made good sense to conduct study in poorly covered and well covered areas, ultimately we compromised on generalizability in favour of practicality. This has been added in limitations (Lines 383-385).

Our baseline survey (data not shown) showed 50% people visited private practitioners, 29% public sector hospitals and 18% traditional medicine.

The last objective is:

8. To compare clinical outcomes (treatment completion, treatment failure, hypoxemia, duration of hospital) of children 0-59 months admitted with severe pneumonia who accepted hospital referral to those who refused admission and were treated at home.

It is difficult to understand this objective as the outcome measure is dependent on hospital admission. It could be rephrased to compare the outcomes of those accepting referrals compared with those being managed elsewhere.

This has been rephrased (Line 116-118)

9. I suspect that those being managed at home would also include children not receiving any treatment. Will there be repeated assessments in the homes after the initial contact?

Yes. The study team will visit all children at day 7 and 14 regardless of their response to hospital referral to collect follow-up data.

VERSION 2 – REVIEW

REVIEWER	Arcott-Mills, Tonya The Children's Hospital of Philadelphia, General Pediatrics
REVIEW RETURNED	29-Apr-2021

GENERAL COMMENTS	I think the issues previously noted have been addressed and I look forward to the results. I hope the funders do allow an extension of the project such that the required number of participants for the power needed are reached.
--

REVIEWER	Lindtjørn , Bernt University of Bergen, Centre for International Health
REVIEW RETURNED	21-Apr-2021

GENERAL COMMENTS	My concern have been addressed
--------------------------------